# CD28 Autonomous Signaling Up-Regulates C-Myc Expression and Promotes Glycolysis Enabling Inflammatory T Cell Responses in Multiple Sclerosis

**DOI:** 10.3390/cells8060575

**Published:** 2019-06-11

**Authors:** Martina Kunkl, Manolo Sambucci, Serena Ruggieri, Carola Amormino, Carla Tortorella, Claudio Gasperini, Luca Battistini, Loretta Tuosto

**Affiliations:** 1Department of Biology and Biotechnology Charles Darwin, Sapienza University, 00185 Rome, Italy; martina.kunkl@uniroma1.it (M.K.); carolaamormino@gmail.com (C.A.); 2Istituto Pasteur Italia-Fondazione Cenci Bolognetti, Sapienza University, 00185 Rome, Italy; 3Neuroimmunology Unit, IRCCS Santa Lucia Foundation, 00185 Rome, Italy; m.sambucci@hsantalucia.it (M.S.); l.battistini@hsantalucia.it (L.B.); 4Department of Neurosciences, S. Camillo/Forlanini Hospital, 00185 Rome, Italy; serena.ruggieri@gmail.com (S.R.); carla.tortorella@gmail.com (C.T.); c.gasperini@libero.it (C.G.)

**Keywords:** multiple sclerosis, inflammation, Th17 cells, glycolysis, CD28, c-myc, class 1A PI3K

## Abstract

The immunopathogenesis of multiple sclerosis (MS) depend on the expansion of specific inflammatory T cell subsets, which are key effectors of tissue damage and demyelination. Emerging studies evidence that a reprogramming of T cell metabolism may occur in MS, thus the identification of stimulatory molecules and associated signaling pathways coordinating the metabolic processes that amplify T cell inflammation in MS is pivotal. Here, we characterized the involvement of the cluster of differentiation (CD)28 and associated signaling mediators in the modulation of the metabolic programs regulating pro-inflammatory T cell functions in relapsing-remitting MS (RRMS) patients. We show that CD28 up-regulates glycolysis independent of the T cell receptor (TCR) engagement by promoting the increase of c-myc and the glucose transporter, Glut1, in RRMS CD4^+^ T cells. The increase of glycolysis induced by CD28 was important for the expression of inflammatory cytokines related to T helper (Th)17 cells, as demonstrated by the strong inhibition exerted by impairing the glycolytic pathway. Finally, we identified the class 1A phosphatidylinositol 3-kinase (PI3K) as the critical signaling mediator of CD28 that regulates cell metabolism and amplify specific inflammatory T cell phenotypes in MS.

## 1. Introduction

Multiple sclerosis (MS) is an autoimmune demyelinating disorder of the central nervous system (CNS) that is characterized by the infiltration of inflammatory immune cells within the CNS [1]. In particular, self-reactive T cells in MS patients cross the blood–brain barrier (BBB) and cause inflammation in the CNS, thus inducing demyelination and disease progression [2]. Studies in both experimental autoimmune encephalomyelitis (EAE) mice [2] and human MS [3,4] evidenced a pivotal role of T helper (Th)17 cells in the pathogenesis of MS [5]. High levels of Th17 cells have been found in the peripheral blood and cerebrospinal fluid (CSF) of patients with relapsing-remitting (RR)MS during relapses [6,7,8] and in active MS patients [9]. Moreover, high levels of Th17 cells have been found in patients with higher number of active plaques, as revealed by magnetic resonance imaging [10]. More recent data evidenced that Th17 cells also produce IL-22, a cytokine that together with other cytokines of the Th17 signature has been implicated in several autoimmune and inflammatory diseases [11,12]. Higher concentrations of IL-22 were detected in the serum of RRMS than in healthy donors [13,14,15,16] and, similarly to Th17 cells, increased IL-22-producing cell number was found in the peripheral blood and CSF of active RRMS patients [17].

Several studies have evidenced a key role of cellular metabolism in regulating the activation, differentiation and functions of T cells [18,19]. In particular, the glucose metabolism controls the proliferation, differentiation and inflammatory functions of T lymphocytes [20]. Specific metabolic pathways have been also described to control the differentiation and inflammatory functions of T cells in autoimmune diseases [21,22]. Elevated glycolysis has been, indeed, associated with Th17 differentiation and functions [23,24,25] and impaired glycolysis has been also shown to correspond with altered suppressive activity of regulatory T lymphocytes (Treg) in MS [26]. The expansion of Th17 cells and reduction of the suppression function and number of Treg represent the main abnormalities in MS [22,27]. Thus, the characterization of receptor molecules and associated signaling mediators, which coordinate the metabolic processes involved in both amplification and/or maintenance of inflammatory T cells in MS, may help in developing novel therapeutic opportunities.

CD28 is a crucial costimulatory receptor for optimal T cell activation that binds B7.1/CD80 or B7.2/CD86 expressed on professional antigen-presenting cells (APC) and augments T cell receptor (TCR) signals required for the production of cytokines, cell proliferation and effector responses [28]. Moreover, human CD28 is also able to deliver individual signals independently of TCR engagement and to induce the transcription and secretion of Th17-related pro-inflammatory cytokines and chemokines [29,30,31] in both healthy subjects [32,33] and in RRMS patients [34]. CD28 contributes to T cell metabolism by enhancing the uptake of nutrients, aerobic glycolysis and anabolic pathways induced by TCR stimulation [22,35]. Indeed, following stimulation by B7 or agonistic antibodies (Abs) the phosphorylated YMNM motif of CD28 intracytoplasmic tail binds the SH2 domain of p85 subunit of class 1A phosphatidylinositol 3-kinase (PI3K) [36] that generates the membrane phospholipids that recruit and activate Akt and mammalian target of rapamycin (mTOR) [37,38,39,40]. The PI3K/Akt/mTOR signaling pathway regulates T cell metabolism by switching metabolism to aerobic glycolysis, promoting Th17 cell differentiation [23,41,42] and inhibiting the proliferation and suppressive functions of Treg cells, thus ameliorating autoimmune EAE [43]. However, the role of CD28 as an individual signaling unit in modulating the metabolic processes regulating T cell inflammatory functions in MS remain still unknown.

In this study, we characterized the role of CD28 individual signaling in the modulation of the metabolic programs regulating pro-inflammatory T cell response in RRMS. We found that CD28 autonomous stimulation up-regulated glycolysis without significantly affecting oxidative phosphorylation in RRMS CD4^+^ T cells. CD28-induced increase of glycolysis was required for the up-regulation of pro-inflammatory cytokines, as demonstrated by the strong inhibition exerted by the glycolysis inhibitor 2-deoxy-D-glucose (2-DG). We also found that CD28-mediated metabolic reprogramming was associated to the increase of c-myc and Glut1. Finally, treatment of CD4^+^ T cells from RRMS patients with a class 1A PI3K inhibitor strongly impaired CD28-induced glycolysis, c-myc and Glut1 expressions, as well as the up-regulation of pro-inflammatory cytokines. These data suggest that CD28 may provide TCR-independent signals modulating the metabolic processes associated to specific pro-inflammatory T cell responses in MS.

## 2. Materials and Methods

### 2.1. MS Patients and Healthy Subjects

Thirty patients with a clinically defined MS according to the McDonald criteria [44] and a clinically RRMS course [45] were enrolled from S. Camillo-Forlanini Hospital (Rome, Italy). The Ethical Committee of S. Camillo-Forlanini had approved the study (ethical code N. 1044/CE Lazio 1, 05/12/2017). All patients signed an informed consent form before blood withdrawal. Patients were between 28 and 70 years old (47.06 ± 10.2), had a disease duration from 1 to 33 years (12.8 ± 10.5) and a mild neurological disability (EDSS, 0.64 ± 1.1). Patients with concomitant severe diseases (neoplasm, respiratory, renal, liver or cardiac failure), recurrent urinary or pulmonary infections or pregnant women were excluded. None of the patients had been treated with steroids or immunosuppressive agents. Most patients were in a stable phase of their disease (no relapse in the last year). Two patients experienced a clinical relapse within the last year. 15 healthy donor (HD) subjects and 15 HD buffy coats from the blood bank of Sapienza University (Rome, Italy) with no previous history of neurological or autoimmune diseases were used as controls.

### 2.2. Cells Abs and Reagents

Human primary CD4^+^ T cells were enriched from PBMC from HD or RRMS patients by negative selection using an EasySepTM isolation kit (STEMCELL Technology, Cambridge, UK) and cultured in RPMI 1640 supplemented with 5% human serum (Euroclone, Milano, Italy), L-glutamine, penicillin and streptomycin. The purity of the sorted population was consistently >95%. The following antibodies were used: Mouse anti-human CD28 (CD28.2, 2 µg mL^−1^), mouse anti-human CD3 (UCHT1, 2 µg mL^−1^), goat anti-mouse (GAM, 2 µg mL^−1^), anti-human CD3-PE (1:10 dilution), anti-human CD4-APC (1:10 dilution) (BD Biosciences); rabbit anti-human Lck (2 µg/chromatin immunoprecipitation (ChIP)), rabbit anti-human RNA polymerase II (N-20, 2 µg/ChIP), rabbit anti-human GAPDH (1:400 dilution; Santa Cruz Biotechnology); mouse anti-human phosphoTyr705 STAT3 (1:1000 dilution for ChIP), rabbit anti-human RelA (1:100 dilution for ChIP; Cell Signaling Technology, Leiden, The Netherlands); mouse anti-human IL-6 (1 µg mL^−1^, R&D Systems, Minneapolis, USA) and mouse anti-human c-myc (Sigma Aldrich, Merck, Milano, Italy). The following inhibitory drugs were used: MG132 (Cayman Chemica, Michigan, USA), PS1145 (Sigma Aldrich, Merck, Milano, Italy), S31-201 (Santa Cruz Biotechnology), AS605240 (Sigma Aldrich, Merck, Milano, Italy), oligomycin (Sigma Aldrich, Merck, Milano, Italy) and 2 deoxy-D-glucose (2-DG, Sigma Aldrich, Merck, Milano, Italy).

### 2.3. Cell Stimulation and Western Blotting

Primary CD4^+^ T cells were stimulated as indicated at 37 °C. At the end of incubation, total cell extracts were obtained by lysing cells for 30 min on ice in 1% Nonidet P-40 lysis buffer (150 mM NaCl, 20 mM Tris-HCl (pH 7.5), 1 mM EGTA, 1 mM MgCl_2_, 50 mM NaF and 10 mM Na_4_P_2_O_7_) in the presence of inhibitors of proteases and phosphatases (10 µg mL^−1^ leupeptin, 10 µg mL^−1^ aprotinin, 1 mM NaVO_4_ and 1 mM pefablock-SC). Proteins were resolved by SDS-PAGE and blotted onto nitrocellulose membranes. Blots were incubated with mouse anti-human c-myc, or mouse anti-human phosphoTyr705 STAT3, or rabbit anti-human GAPDH primary antibodies, extensively washed and after incubation with horseradish peroxidase (HRP)-labeled goat anti-rabbit (1:5000 dilution) or HRP-labeled goat anti-mouse (1:5000 dilution; GE Healthcare, Chicago, IL, USA) developed with the enhanced chemiluminescence’s detection system (GE Healthcare, Chicago, IL, USA). Protein levels were quantified by densitometric analysis using the ImageJ 1.50i program (National Institute of Health (NIH), Bethehda, MD, USA).

### 2.4. Chromatin Immunoprecipitation (ChIP)

ChIP assays were performed as previously described [46]. Briefly, after fixing in 1% formaldehyde, T cells were lysed for 5 min in 50 mM Tris, pH 8.0, 2 mM EDTA, 0.1% NP-40 and 10% glycerol supplemented with proteases inhibitors. Nuclei were suspended in 50 mM Tris, pH 8.0, 1% SDS and 5 mM EDTA. Chromatin was sheared by sonication, centrifuged and diluted 10 times in 50 mM Tris, pH 8.0, 0.5% NP-40, 0.2 M NaCl and 0.5 mM EDTA. After pre-clearing with a 50% suspension salmon sperm-saturated Protein-A or Protein G Sepharose beads (GE Healthcare, Chicago, USA), lysates were incubated at 4 °C overnight with anti-RelA (1:100 dilution), anti phoshoTyr705 STAT3 (pSTAT3, 1:100 dilution), anti-RNA-polymerase II (Pol II, 2 µg) or anti-Lck (2 µg) Ab as control. Immune complexes were collected with sperm-saturated Protein-A or Protein G Sepharose beads, washed three times with a high salt buffer (20 mM Tris, pH 8.0, 0.1% SDS, 1% NP-40, 2 mM EDTA and 500 mM NaCl) and five times with 1× Tris/EDTA (TE). Immune complexes were extracted in 1× TE containing 1% SDS, and protein–DNA cross-links were reverted by heating at 65 °C overnight. DNA was extracted by phenol–chloroform and about 1/30 of the immunoprecipitated DNA was analyzed by real-time PCR. Quantitative real-time PCR with SYBR Green Supermix (Bio-Rad, Milano, Italy) was performed for the human c-myc promoter P2 [47,48,49,50]. Specific enrichment was calculated as previously described [51] by using the cycle threshold (Ct): 2^(Ct of control ChIP – Ct of control input)^/2^(Ct of specific ChIP – Ct of specific input)^. The c-myc promoter primers used for each specific ChIP were as follows: RelA, 5′-GGAACTTACAACACCCGAGCAAGG-3′ and 5′-CCTTTCAGAGAAGCGGGTCCTG-3′; pSTAT3 and Pol II, 5′-CTCCTGCCTCGAGAAGGGCAG-3′ and 5′-CTCCCTCTGCCTCTCGCTGGA-3′.

### 2.5. Multicolour Analysis of Surface Activation Markers

CD4^+^ T cells from RRMS patients were stained with anti-human CD4-FITC, anti-human CD39-APC, anti-human CCR6 PE-Vio615 (Miltenyi Biotech, Bologna, Italy); anti-human CD71-PE, anti-human PD-1-BV650, anti-human Glut1-PE (Becton Dickinson Italia S.p.a, Milano, Italy); anti-human CD25-PE-CF594, anti-human CD69-BV421 (Pharmingen, Becton Dickinson Italia S.p.a, Milano, Italy) and anti-human CD28-PerCP-Cy5.5, anti-human CXCR3 PerCP-Cy5.5 (Biolegend, San Diego, USA). Live/Dead Fixable Aqua Dead cell stain (Invitrogen, Thermo Fisher Scientific, Milano, Italy) was also added to the cocktail for excluding dead cells. Cells were acquired on a Beckman Coulter Cytoflex flow cytometer and analyzed using the FlowJo v10.5.3 software.

### 2.6. Real-Time PCR

The total RNA was extracted using Trizol (Thermo Fisher Scientific, Milano, Italy) from 1 × 10^6^ cells or RNeasy MicroKit (Qiagen Italia, Milano, Italy) from 5 × 10^5^ cells according to the manufacturer’s instructions and was reverse-transcribed into cDNA by using Moloney murine leukemia virus reverse transcriptase (Invitrogen). TaqMan Universal PCR Master Mix and the human primer/probe sets for IL-21 (Hs00222327_m1), IL-17A (Hs00174383_m1), IL-22 (Hs01574154_m1), IL-6 (Hs00985639_m1), GAPDH (Hs99999905_m1), MYC (Hs00153408_m1), HK2 (Hs00606086_m1), ENO1 (Hs00361415_m1), PDK1 (Hs01561847_m1), LDHA (Hs01378790_g1), HIF1α (Hs00153153_m1) and G6PD (Hs00166169_m1) were purchased from Applied Biosystems. The PCR was performed by using the following program: One cycle of 10 min at 95 °C followed by 60 cycles of denaturation for 15 s at 95 °C and an annealing/extension step of 1 min at 60 °C. The relative quantification was performed using the comparative C_T_ method. The mean value of human CD4^+^ T cell stimulated with the control isotype matched Ab was used as the C_T_ calibrator in all comparative analyses.

### 2.7. Real Time Analysis of Cell Metabolism

Real-time measurements of the extracellular acidification rate (ECAR) and oxygen consumption rate (OCR) were performed using an XFe-96 Extracellular Flux Analyzer (Seahorse Bioscience). CD4^+^ T cells from RRMS patients or HD were seeded in XFe-96 plates (Seahorse Bioscience) at the density of 4 × 10^5^ cells/well and stimulated overnight at 37 °C in 5% CO_2_ atmosphere with isotype control Ig or the indicated Abs in the presence or absence of DMSO, as the vehicle control, or AS605240, where indicated. ECAR was measured in an XF medium in basal condition and in response to 10 mM glucose, 5 μM oligomycin and 100 mM of 2-Deoxy-d-glucose (2-DG; all from Sigma Aldrich). Basal glycolysis was calculated after the addition of glucose, maximal glycolysis was measured after the addition of oligomycin and glycolytic capacity as the difference between oligomycin-induced ECAR and 2-DG-induced ECAR. OCR was measured in XF medium under basal conditions and in response to 5 μM oligomycin, 1.5 μM of carbonylcyanide-4-(trifluoromethoxy)-phenylhydrazone (FCCP) and 1 μM of antimycin A and rotenone by using a commercial kit (Seahorse XF Cell Mito Stress Test, Seahorse Bioscience). Indices of mitochondrial respiratory function were calculated from the OCR profile: Basal OCR (before the addition of oligomycin), ATP-linked OCR (calculated as the difference between the basal OCR rate and oligomycin-induced OCR rate), maximal OCR (calculated as the difference of the FCCP rate and antimycin + rotenone rate) and SRC (calculated as the difference between the maximal respiration and basal respiration; spare respiratory capacity is a measure of the ability of the cell to respond to increased energy demand or under stress).

### 2.8. Statistical Analysis

The sample size was chosen based on previous studies to ensure adequate power. Parametrical statistical analysis (mean and SEM) was performed to evaluate differences between continuous variables through Prism 5.0 (GraphPad Software, San Diego, CA) using an unpaired Student’s *t*-test or Wilcoxon test. For multiple group comparisons, significant differences were calculated using the nonparametric Mann–Whitney *U* test, and a linear regression analyses were performed using the Pearson chi-squared test. For all tests, *p* values < 0.05 were considered significant.

## 3. Results

### 3.1. CD28 Pro-Inflammatory Functions Are Associated With a Glycolytic Metabolic Reprogramming

Several studies evidenced the important contribution of CD28 costimulation in regulating TCR-mediated up-regulation of the glycolytic metabolism [22,35]. However, the role of CD28 as a TCR-independent signaling unit in reprogramming the metabolic processes regulating the T cell effector function and oxygen-consumptions remains still unknown. To this aim, CD4^+^ T cells from HD were stimulated with agonistic anti-CD28 (CD28.2) alone or in combination with anti-CD3 (UCHT1) or isotype control Abs and after 18 h aerobic glycolysis and oxidative phosphorylation were analyzed by measuring the extracellular acidification rate (ECAR) and oxygen-consumption rate (OCR), respectively. Following CD28 ligation alone, CD4^+^ T cells switched their metabolic state by up-regulating the aerobic glycolytic flux at levels comparable to anti-CD3 plus anti-CD28 stimulation (Figure 1a). The increase in the glycolytic flux (Figure 1a) and glycolytic capacity (Figure 1c) in response to CD28 was also accompanied by the up-regulation of both basal (Figure 1c) and maximal glycolytic responses (Figure 1d). In contrast, no significant changes in oxidative phosphorylation (OCR, Figure 1e), maximal respiration (Figure 1e) and spare respiratory capacity (SCR, Figure 1g), were observed.

CD28 stimulation induced a significant increase of glycolysis also in age–sex matched stable RRMS patients, who had not undergone any treatment, as demonstrated by the increase of ECAR (Figure 2a), glycolytic capacity (Figure 2b) and maximal glycolysis (Figure 2c) observed in CD4^+^ T cells following stimulation with agonistic anti-CD28 Abs. No significant differences were observed in the up-regulation of glycolysis between RRMS patients and HD following CD28 engagement (Appendix A). As observed in HDs (Figure 1e–g), mitochondrial oxidative phosphorylation did not significantly change in CD28-stimulated CD4^+^ T cells from RRMS (Figure 2d,e). The glycolytic switch induced by CD28 signals was also associated with the increase of surface activation markers, such as CD69, CD71 and CD25 (Appendix A), whereas the expression of PD-1 was not modulated (Appendix A). Consistently with our previous data [34], the increase of glycolysis was also associated with the increase of transcription of Th17 cell-related pro-inflammatory cytokines, such as IL-6, IL-21, IL-22 and IL-17A (Appendix A). More importantly, a strong increase of the glucose transporter Glut1 was also detected on the surface of RRMS CD4^+^ T cells following CD28 engagement (Appendix A). Interestingly, CD4^+^ T cells from RRMS expressed significant higher baseline levels of Glut1 compared to HD subjects and CD28 stimulation induced a higher Glut1 up-regulation in RRMS patients (Appendix A). Moreover, the analysis of Glut1 expression on different T cell subsets in RRMS patients, revealed that Th17-like cells (CXCR3^−^CCR6^+^) expressed higher levels of Glut1 compared to Th1-like cells (CXCR3^+^CCR6^−^) or Th0-like cell (CXCR3^−^CCR6^−^) and CD28 stimulation up-regulated Glut1 expression in all T cell subsets, although to a higher extent in Th17 cells (Appendix A).

To assess the relevance of the glycolytic metabolism in CD28 pro-inflammatory functions, CD4^+^ T cells from RRMS were activated in the presence or absence of 2-DG, a selective inhibitor of glycolysis. The presence of 2-DG strongly impaired CD28-induced up-regulation of pro-inflammatory cytokine gene expression in RRMS CD4^+^ T cells, thus supporting the pivotal role of glycolysis in CD28 pro-inflammatory function (Figure 2f–i).

### 3.2. CD28-Induced Metabolic Reprogramming is Related to the Increase of C-Myc

To examine the mechanisms of CD28-induced metabolic reprogramming, we analyzed the expression of the major enzymes regulating glycolysis, such as hexokinase 2 (HK2), enolase 1 (ENO1), pyruvate dehydrogenase kinase 1 (PDK1), glucose 6 phosphate dehydrogenase (G6PD), HIF-1 and lactate dehydrogenase (LDHA) [52]. No significant increase in the mRNA levels of HK2 (Figure 3a), ENO1 (Figure 3b), PDK1 (Figure 3c) or G6PD (Figure 3d) were observed in T cells from neither HD nor RRMS following stimulation with agonistic anti-CD28 Abs. Notably, a significant up-regulation of LDHA (Figure 3e) and HIF-1 (Figure 3f) gene expression was observed in HD and/or RRMS patients. However, the immunoblot analysis did not confirm the up-regulation of these metabolic enzymes at the protein level (Figure 3g,h).

C-myc is another important regulator of glucose uptake and T cell metabolism [53,54]. We therefore analyzed both gene and protein expression of c-myc in CD28-stimulated CD4^+^ T lymphocytes from HDs. CD28 stimulation strongly increased c-myc gene expression within 3 h followed to a decrease after 6 h, and returning to a basal level after 24 h (Figure 4a). The protein expression studies confirmed the data obtained from mRNA analysis with a peak of c-myc expression at 3 h and a decrease at a basal level after 24 h (Figure 4b, upper panels and lower graph). CD28 stimulation increased c-myc protein expression also in CD4^+^ T cells from RRMS and not significant differences in c-myc expression were observed between RRMS and HD (Figure 4c).

We next analyzed the mechanisms and transcription factors involved in CD28-mediated c-myc up-regulation. The human c-myc gene contains four promoters (P0–P3) and in 75–90% normal cells c-myc transcription initiates at the level of promoter P2 [55]. The P2 promoter of c-myc contains a specific binding site for active phosphorylated STAT3 (TTGGCGGGAA) that binds and trans-activates c-myc in response to IL-6 [47,48]. Moreover, human c-myc gene also contains two specific binding sites for RelA/NF-κB and one (GGGACACTTC) is located within the regulatory internal element (IRE) in exon 1 very close to the STAT3 binding site of promoter P2 and the relative starting site [49,50] (Figure 4d). We have previously demonstrated that CD28 engagement promoted the activation and translocation to the nucleus of RelA/NF-κB [29]. More recently, we showed that CD28 induces the phosphorylation on Tyr705 of STAT3 (pSTAT3) and its nuclear translocation in an IL-6-dependent manner in HD subjects [32] and in RRMS patients (Appendix A). In CD4^+^ T cells from HD subjects, ChIP experiments revealed a strong increase of both pSTAT3 (Figure 4e) and RelA (Figure 4f) binding to the c-myc promoter after 3 h of stimulation with anti-CD28.2 Ab. The binding of both transcription factors to the c-myc promoter was also accompanied by its transcriptional activation, as demonstrated by the recruitment of RNA polymerase II (pol II; Figure 4g). Interestingly, in CD4^+^ T cells from RRMS patients, the addition of anti-IL-6 neutralizing Abs strongly impaired c-myc protein up-regulation induced by CD28 engagement (Figure 4h,i). Moreover, treatment of CD4^+^ T cells from RRMS with S31-201, a selective STAT3 inhibitor, (Figure 4l) as well as PS1145, an NF-κB inhibitor, (Figure 4m) strongly impaired CD28-induced c-myc expression. Similar results were obtained in HD subjects (Appendix A), thus suggesting a cooperation of IL-6-mediated STAT3 activation and RelA/NF-κB in CD28-mediated up-regulation of c-myc expression in both HD and RRMS patients.

### 3.3. Class 1A PI3K Is Required for CD28-Induced Metabolic Reprogramming and Inflammatory Functions of RRMS T Lymphocytes

Several studies evidenced a key role of PI3K/PDK1/Akt and mTOR pathways in the metabolic switch toward aerobic glycolysis, thus leading to the differentiation of specific effector T helper cells. In particular, PI3K/PDK1/Akt/mTOR pathway drives Th17 differentiation [23,41], while the impairment of mTOR activity increases the expansion of Treg cells and their suppressive functions, thus ameliorating EAE [43]. In T cells, CD28 is the prominent inducer of PI3K/PDK1/Akt/mTOR through the recruitment and activation of class 1A PI3K [36,37,38,56]. Peripheral CD4^+^ T lymphocytes from RRMS patients were activated with anti-CD28.2 Abs with or without AS605240 class 1 PI3K inhibitor and the level of glycolysis (Figure 5a,b), the expression of Glut1 (Figure 5c) and of activation markers (Figure 5d), pro-inflammatory cytokine expression (Figure 5e–h) as well as c-myc (Figure 5i,l) were analyzed. Overall the inhibition of class 1A PI3K by AS605240 significantly impaired all the functional activities analyzed, thus evidencing that class 1A PI3K is a critical mediator of CD28 signals, which regulate the metabolism of peripheral inflammatory T lymphocytes MS.

## 4. Discussion

The differentiation and amplification of pro-inflammatory T cells has been extensively correlated with a switch of the metabolism to anabolic glycolysis [57] and the impairment of glycolysis as well as the accumulation of specific metabolites in the brain have been recently related with a poor prognosis in MS patients [58]. Therefore, the characterization of the mechanisms and molecules, which modulate the metabolic processes regulating T cell-mediated inflammation could be important for the design of more efficient therapeutic interventions in MS. For instance, several first-line therapies used in MS, although effective in suppressing self-reactive T lymphocytes, are not efficient in progressive phases of the disease [59]. Second generation drugs, although exerting potent anti-inflammatory activity, may induce severe secondary effects [60]. Here, we show that CD28 individual signaling induces glycolysis enabling inflammatory T cell functions in RRMS patients.

The contribution of CD28 in regulating the metabolic programs associated to T cell effector functions [35,61,62,63] has been always analyzed together with TCR that has been also recently described to initiate the signaling events required for aerobic glycolysis [64]. However, human CD28 is also able to activate signals independently of TCR and to amplify pro-inflammatory T cell responses by inducing the expression and secretion of Th17-related cytokines [30,32,34]. Th17 cells contribute to neuroinflammation in MS by secreting several effector cytokines, including IL-17, whose levels are higher in the CSF during the exacerbation of the disease [6,7,8,9,10]. Pro-inflammatory Th17 cells also utilizes a high rate of glycolysis for sustaining their energy and biosynthetic needs [25,56] and recent data from Shi et al. evidenced a decrease of IL-17 production and EAE disease severity in mice treated with the prototypical inhibitor of glycolysis, 2-DG [23]. Consistently with these data, we found that, in RRMS patients, CD28-mediated up-regulation of Th17 cytokines (Appendix A) was associated to an increase of the glycolytic program (Figure 2a–c) and was strongly impaired by 2-DG glycolysis inhibitor (Figure 2g–l). In contrast to other studies, showing either an impairment [65] or an increase of glycolysis [66] in RRMS patients when T cells were stimulated by TCR, no significant differences in the strength of the glycolytic flux were observed between RRMS and HD after CD28 stimulation (Appendix A), thus suggesting that, when engaged independently of TCR, CD28 may activate distinct transcriptional and/or translational changes promoting the glycolytic switch required for sustaining inflammatory T cell functions.

The up-regulation of Glut1 glucose transporter is required for both glycolysis and the effector functions of T lymphocytes [35,67]. For instance, high levels of Glut1 expression have been related to higher glycolytic activity and functions of distinct effector CD4^+^ T cell subset [25] and the PI3K/Akt/mTOR pathway has been implicated in both the expression and trafficking of Glut1 [35,68]. Moreover, higher levels of Glut1 have been recently observed by De Riccardis et al. in MS CD4^+^ T lymphocytes [66]. Likewise, in RRMS patients, we found that CD4^+^ T lymphocytes expressed higher basal levels of Glut1 than HD (Appendix A). CD28 autonomous stimulation up-regulated Glut1 expression in CD4^+^ T cells in RRMS patients (Appendix A), at higher levels than HD (Appendix Ac) and, in particular, on the inflammatory Th17 cell subset (Appendix A). In agreement with the main role of CD28 in activating the PI3K/AKT/mTOR pathway [34,38,69], we found that CD28-induced up-regulation of Glut1 was inhibited by AS605240 class 1A PI3K inhibitor (Figure 5c). The increase of Glut1 induced by CD28 stimulation was not accompanied by a transcriptional up-regulation of the major enzymes (Figure 3), which have been correlated with a glycolytic metabolic switch in both conventional [70] and effector/memory T lymphocytes [71], but was associated with a strong increase of c-myc (Figure 4).

In T lymphocytes, c-myc is a critical regulator of cell metabolism and a potential downstream target of mTOR [53,72]. In the mouse system, deletion of c-myc strongly impairs the metabolic switch to glycolysis induced by TCR and CD28 co-engagement [54]. Moreover, c-myc expression is also essential for Th17 differentiation [53] and c-myc suppression by Foxp3 in Treg cells has been also associated to the impairment of glycolysis and enhancement of oxidative phosphorylation [73]. Interestingly, we found that CD28 signaling through the activation of PI3K rapidly induced c-myc expression in RRMS patients (Figure 4, Figure 5). We also demonstrated that CD28-induced c-myc expression depended on IL-6 that, as we have previously demonstrated [32], is strongly up-regulated in both HD and RRMS patients by CD28 (Appendix A) and by activating STAT3 in both HD [32] and RRMS patients (Appendix A) cooperated with RelA/NF-κB, as demonstrated by the strong inhibition of c-myc expression exerted by culturing T cells in the presence of anti-IL-6 Abs (Figure 4i) or S31-201 STAT3 inhibitor, (Figure 4l) or PS1145 NF-κB inhibitor (Figure 4m). Interestingly, STAT3 and RelA/NF-κB are important transcription factors for both c-myc [47,48,49,50] and IL-17A [74,75,76] and we have recently demonstrated that STAT3 cooperates with RelA/NF-κB in trans-activating the human IL-17A gene promoter [32]. Similarly, STAT3 and RelA/NF-κB were earliest recruited on the human c-myc promoter (Figure 4d) in response to CD28 and mutually cooperated for inducing its transcriptional activation (Figure 4e–g). Finally, the impairment of the glycolytic switch, pro-inflammatory cytokines and c-myc expression exerted by a selective inhibitor (Figure 5) evidenced a critical role of class 1A PI3K in regulating the glycolytic program of inflammatory T lymphocytes in MS.

## 5. Conclusions

Recently a single nucleotide polymorphism (rs4410871) in Myc has been identified as a risk allele linked to human MS [77]. This discovery together with our data on the critical role of the CD28/PI3K/c-myc axis in regulating the glycolytic metabolism of inflammatory T lymphocytes may provide the bases for the development of more efficient immunotherapeutic programs that target CD28 and associated signaling molecule in order to suppress excessive inflammation in MS.

## Figures and Tables

**Figure 1 cells-08-00575-f001:**
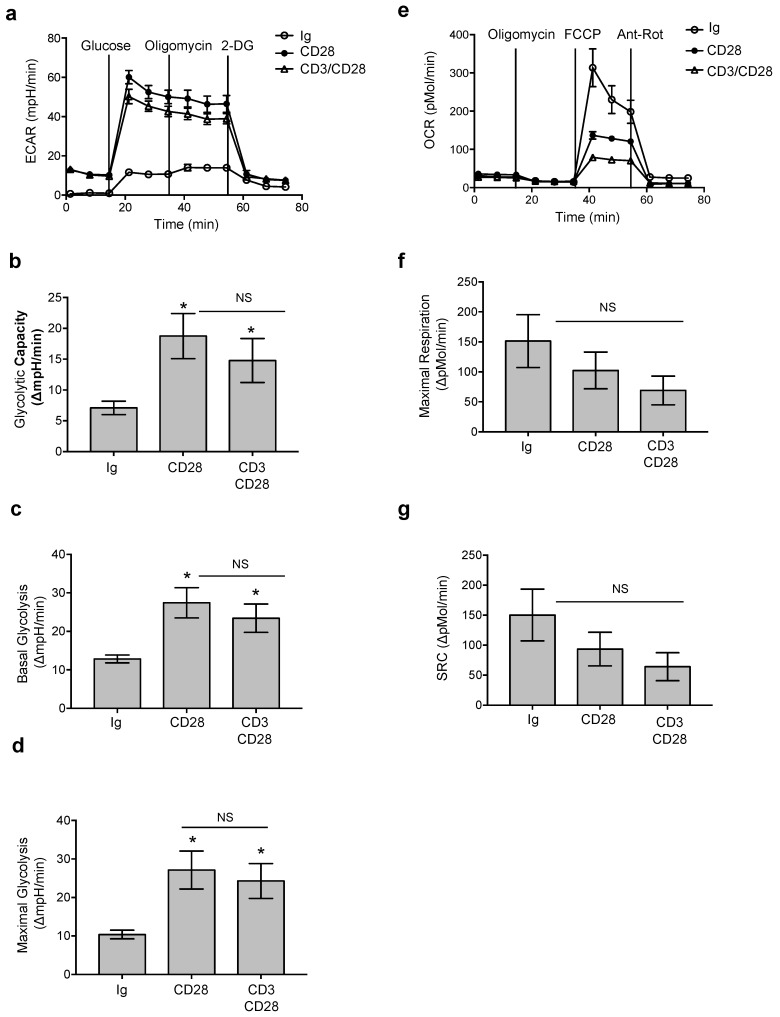
CD28 activates glycolysis in CD4^+^ T cells. (**a**) Peripheral blood CD4^+^ T cells from a representative healthy donor (HD) were stimulated for 18 h with 2 µg mL^−1^ isotype control Ig, or anti-CD28.2 or anti-CD28.2 plus anti-CD3 (UCHT1) Abs. The kinetic profile of the extracellular acidification rate (ECAR), was measured by Seahorse analysis, at a basal level and after addition of glucose, oligomycin and 2-DG. Data express the mean ± SEM of sextuplicate cultures. (**b–d**) CD4^+^ T cells from HDs (*n* = 7) were activated as in (**a**) and glycolytic capacity (**b**), basal glycolysis after glucose injection (**c**) and maximal glycolysis (**d**) were calculated from the ECAR profiles. Data express mean ± SEM. (**e**) CD4^+^ T cells from a representative HD were activated as in (**a**) and the oxygen consumption rate (OCR) was measured by Seahorse analysis at a basal level and after addition of oligomycin, FCCP, antimycin A and rotenone (Ant-Rot). Data express the mean ± SEM of sextuplicate cultures. (**f**,**g**) Maximal respiration (**f**) and spare respiratory capacity (SRC) of CD4^+^ T cells from HDs (*n* = 5) activated as in (**a**) were calculated from the OCR profiles. Data express mean ± SEM and significance was calculated by Wilcoxon test. (*) *p* < 0.05, NS = not significant.

**Figure 2 cells-08-00575-f002:**
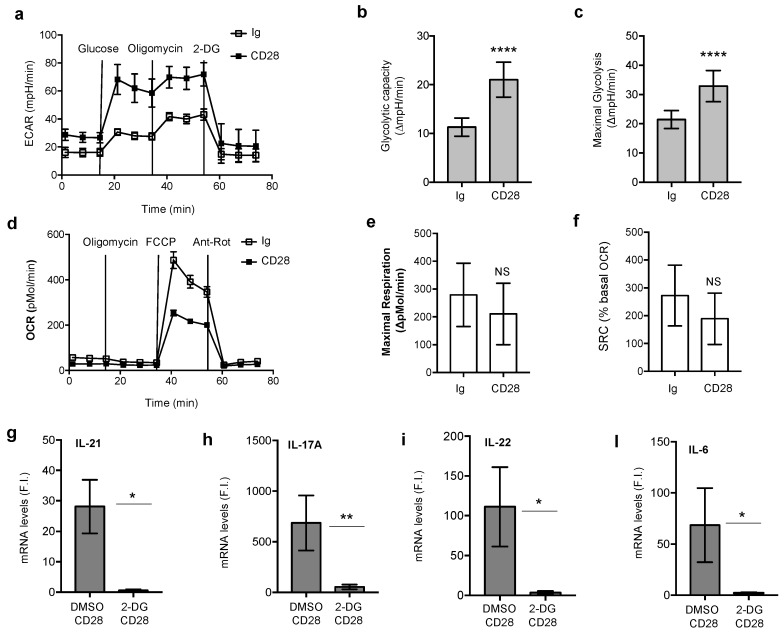
CD28 inflammatory functions in CD4^+^ T cells from relapsing-remitting multiple sclerosis (RRMS) patients depends on glycolysis. (**a**) Peripheral blood CD4^+^ T cells from a representative RRMS patient were activated for 18 h with 2 µg mL^−1^ isotype control Ig or anti-CD28.2 Abs. The kinetic profile of ECAR was measured at basal levels and following the addition of glucose, oligomycin or 2-DG. Data express the mean ± SEM of sextuplicate cultures. (**b,c**) CD4^+^ T cells from RRMS (*n* = 17) were stimulated as in (**a**) and glycolytic capacity (**b**) and maximal glycolysis (**c**) were calculated from the ECAR profiles. Data express the mean ± SEM. (**d**) Peripheral blood CD4^+^ T cells from a representative RRMS patient were activated as in (**a**) and OCR was measured at basal levels and following the addition of oligomycin, FCCP or antimycin A and rotenone (Ant-Rot). Data express the mean ± SEM of sextuplicate cultures. (**e,f**) CD4^+^ T cells from RRMS (*n* = 3) were activated as in (**a**) and maximal respiration (**e**) and SRC (**f**) were calculated from the OCR profiles. Data express the mean ± SEM. (****) *p* < 0.0001 by Wilcoxon test. NS = not significant. (**g–l**) CD4^+^ T cells from RRMS (*n* = 10) cultured with DMSO, as control, or 100 µM 2-DG were activated for 6 h (**g,l**) or 24 h (**h,i**) with 2 µg mL^−1^ isotype control Ig or anti-CD28.2 Abs and IL-21 (**g**), IL-17A (**h**), IL-22 (**i**) and IL-6 (**l**) mRNA levels were analyzed by real time PCR. The values expressed as fold inductions (F.I.) over isotype control Ig-stimulated cells after normalization to GAPDH (**e**). Bars represent the mean ± SEM and statistical significance was calculated by an unpaired Student’s *t* test. (*) *p* < 0.05, (**) *p* < 0.01.

**Figure 3 cells-08-00575-f003:**
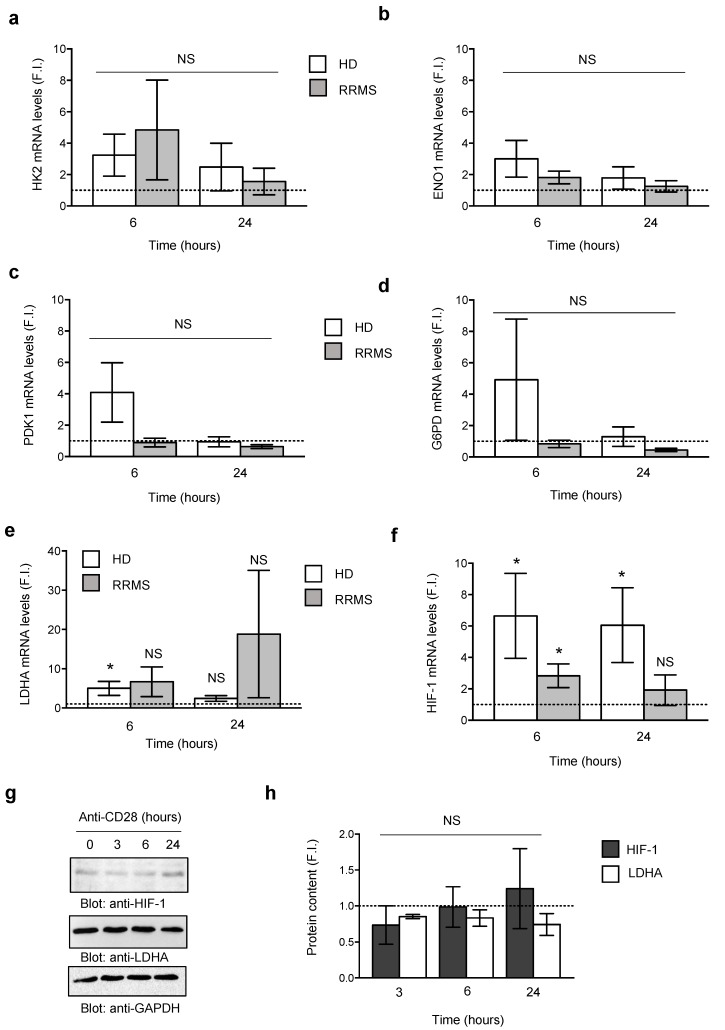
Metabolic profile of CD4^+^ T cells from HDs or RRMS. (**a–f**) Peripheral blood CD4^+^ T cells from HD or RRMS patients were activated for 6 or 24 h with 2 µg mL^−1^ isotype control Ig or anti-CD28.2 Abs and mRNA levels of hexokinase 2 (HK2, *n* = 5), enolase 1 (ENO1, *n* = 6), pyruvate dehydrogenase kinase 1 (PDK1, *n* = 6), glucose 6-phosphate dehydrogenase (G6PD, *n* = 6), lactate dehydrogenase (LDHA, *n* = 13) and hypoxia factor 1 (HIF-1, *n* = 13) were measured by real time PCR. The values were expressed as fold inductions (F.I.) over isotype control Ig-stimulated cells after normalization to GAPDH. Data express the mean ± SEM. (**g**,**h**) HIF-1, LDHA and GAPDH western blotting of CD4^+^ T cells from HD activated for the indicated times with anti-CD28.2 Abs. (**h**) Fold inductions (F.I.) over isotype control Ig-stimulated cells were quantified by densitometric analysis and normalized to GAPDH levels. Bars represent mean F.I. ± SEM of three HD. Statistical significance was calculated by an unpaired Student’s *t* test. (*) *p* < 0.05. NS = not significant.

**Figure 4 cells-08-00575-f004:**
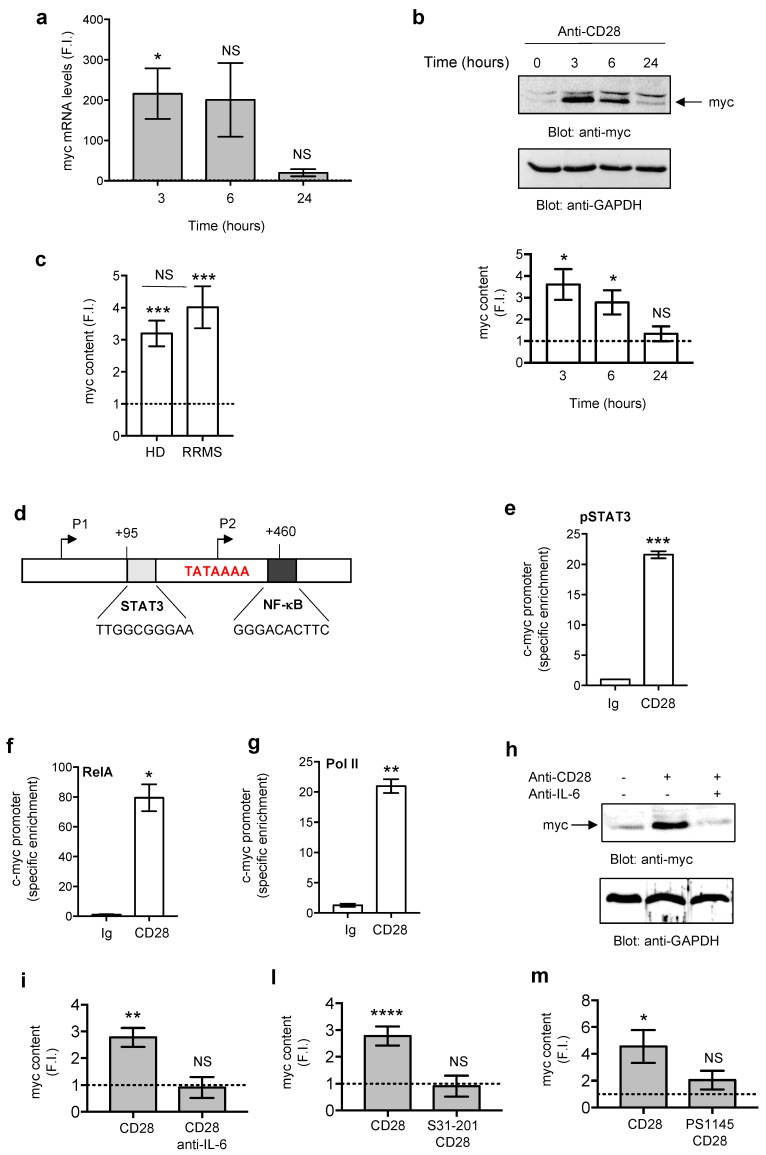
STAT3 and RelA/NF-κB transcription factors regulate the transactivation of c-myc in response to CD28 stimulation. (**a**) Peripheral blood CD4^+^ T cells from HDs (*n* = 3) were activated for 3, 6 or 24 h with isotype control Ig or anti-CD28.2 Abs and c-myc mRNA levels were measured by real time PCR. The values were expressed as fold induction (F.I.) over isotype control Ig after normalization to GAPDH. Data show the mean ± SEM. (**b**) Western blotting analysis was used to measure c-myc and GAPDH levels in CD4^+^ T cells from HDs (*n* = 3) activated for different times with anti-CD28.2 Abs or isotype control Ig. Fold inductions (F.I.) were calculated after normalization to GAPDH levels and expressed as mean F.I. ± SEM (lower graph). (**c**) Anti-myc western blotting of CD4^+^ T cells from HDs (*n* = 6) or RRMS (*n* = 6) stimulated for 3 h with anti-CD28.2 Abs or isotype control Ig. Fold inductions (F.I.) were calculated after normalization to GAPDH levels and expressed as mean F.I. ± SEM. (**d**) Schematic sequence of the human c-myc P2 promoter gene with TATA box (red) and the specific binding sites for NF-κB and STAT3. (**e–f**) Anti-pSTAT3 (**e**), anti-RelA (**f**) and anti-Poll II (**g**) ChIP of CD4^+^ T cells activated for 3 h with isotype control Ig or anti-CD28.2 Abs. Real time PCR with primers specific for the c-myc P2 promoter were performed and the specific enrichment was calculated by the Cτ method. Data show the mean ± SEM of one representative out of three HDs. (**h–m**) CD4^+^ T cells from RRMS patients (*n* = 3) were activated as in (**g**) in the presence or absence of 10 µg mL^−1^ neutralizing anti-IL-6 Abs (**h,i**), or 100 µM S31-201 (**l**), or 10 µM PS1145 (**m**) and anti-myc and anti-GAPDH western blotting were performed. Fold inductions (F.I.) over isotype control Ig were calculated after normalization to GAPDH levels. Bars show mean F.I. ± SEM and statistical significance was calculated by an unpaired Student’s *t* test. (*) *p* < 0.05, (**) *p* < 0.01, (***) *p* < s0.001, (****) *p* < 0.0001. NS = not significant.

**Figure 5 cells-08-00575-f005:**
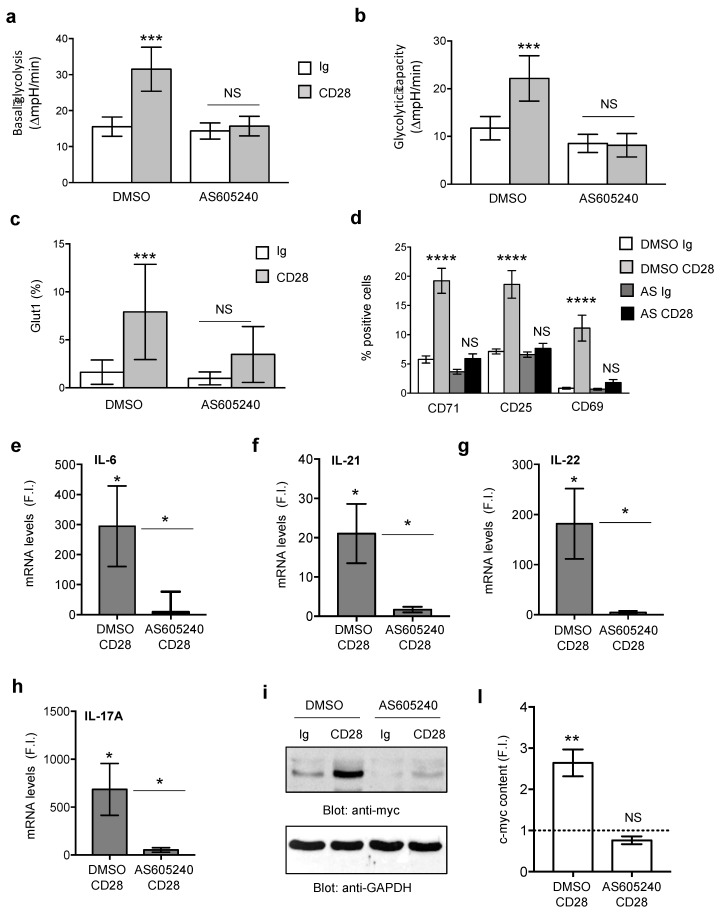
CD28-mediated metabolic reprogramming and pro-inflammatory functions of CD4^+^ T cells from RRMS patients depend on class 1A PI3K activity. (**a**,**b**) CD4^+^ T cells from RRMS (*n* = 13) were stimulated for 18 h with isotype control Ig or anti-CD28.2 Abs in the presence of DMSO, as vehicle control, or 10 µM AS605240. Basal glycolysis after glucose injection (**a**) and maximal glycolysis (**b**) were calculated from the ECAR profiles. Data express the mean ± SEM. Significance was calculated by Wilcoxon test. (**c**,**d**) Multicolor flow cytometry analysis of Glut1 (**c**), CD71, CD69 and CD25 (**d**) on CD4^+^ T cells from RRMS patients (*n* = 17) stimulated for 24 h as in (**a**). Data express the mean ± SEM. (**e–h**) Real time PCR of the indicated cytokines in CD4^+^ T cells from RRMS patients (*n* = 15) stimulated for 6 h (**e**,**f**) or 24 h (**g**,**h**) as in (**a**). After normalization to GAPDH levels, fold induction (F.I.) over DMSO-treated isotype control Ig was calculated. Data show the mean ± SEM. (**i**,**l**) Anti-c-myc (upper panel) and anti-GAPDH (lower panel) western blotting of CD4^+^ T cells from RRMS (*n* = 3) stimulated for 3 h as in (**a**). (**l**) Fold inductions (F.I.) over isotype control Ig were calculated after normalization to GAPDH levels. Bars show the mean F.I. ± SEM and statistical significance was calculated by an unpaired Student’s *t* test. (*) *p* < 0.05, (**) *p* < 0.01, (***) *p* < 0.001, (****) *p* < 0.0001. NS = not significant.

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
