# Peer review of "CD28 Autonomous Signaling Up-Regulates C-Myc Expression and Promotes Glycolysis Enabling Inflammatory T Cell Responses in Multiple Sclerosis"

_cells, 2019, doi:10.3390/cells8060575_

Round 1

Reviewer 1 Report

Comments:

In the current study, authors showed that CD28 up-regulates glycolysis independently of T cell receptor engagement by promoting the increase of c-myc and the glucose transporter, Glut1, in CD4+ cells from RRMS patients. Also, authors further revealed that CD28-induced increase of glycolysis was also associated with the up-regulation of Th17 related inflammatory cytokines, as verified by the strong inhibition exerted by impairing the glycolytic pathway. In conclusion, authors identified CD28-associated class 1A phosphatidylinositol 3-kinase as the critical signalling mediator of the metabolic processes that amplify the inflammatory phenotype of peripheral T cells in MS. It is an interesting study and results are clearly presented. The manuscript is easy to read and understand.

Minor comments

Page 1, Line 24: Abbreviate the CD28

Page 1, Line 24: Provide the Proper abbreviation for RRMS

Page 3 section 2.3: Include the protein names in this section

Page 3 section 2.6: Authors can include the primer sequences in 2.6 section also mentioned the details of the Real-time PCR cycles.

Page 8: Authors needs to increase the resolution of figure 2

Author Response

We thank the reviewer for suggestions and advices. To better clarify our responses, we numbered the reviewer’s comments and our responses are indicated below

Page 1, Line 24: Abbreviate the CD28

Response: CD28 has been abbreviated.

Page 1, Line 24: Provide the Proper abbreviation for RRMS

Response: The proper abbreviation for RRMS has been added

Page 3 section 2.3: Include the protein names in this section

Response: The names of the proteins that were analysed by western blotting have been added

Page 3 section 2.6: Authors can include the primer sequences in 2.6 section also mentioned the details of the Real-time PCR cycles.

Response: All oligonucleotides used for gene expression analysis have been purchased by Applied Biosystems that does not provide their sequence. However, we added the specific code of each primer set and details of the real-time PCR cycles, as required

Page 8: Authors needs to increase the resolution of figure 2

Response: The resolution of Figure 2 has been increased

Reviewer 2 Report

Overall, the paper is very well written by a group known for studying CD28 signaling. The data and conclusions are interesting. However, there are issues with some experiments and data presented here.  The authors stated that the blockade of IL-6 mediated signaling inhibits CD28-induced c-myc expression in CD4 T cells from RRMS patients.  In the present form is not clear if there are any significant differences  in c-myc up regulation between  T cell from HD and MS. Again, in Fig. 4 (panel h-m) samples from HD treated as MS samples including the treatment with anti-IL6 mAb should be added.  Quantitation of IL-6 induced by anti-CD28 treatment in both HD and MS should be also measured. Is there any difference in IL-6 secretion between HD and MS under this condition? Analysis of STAT3 and NFKB activation in MS and HD following IL-6 or anti-CD28 treatment should be included in figure 4 or in a supplementary figure. 

Author Response

We thank the reviewer for suggestions and advices. To better clarify our responses, we numbered the reviewer’s comments and our responses are indicated below

1. The authors stated that the blockade of IL-6 mediated signalling inhibits CD28-induced c-myc expression in CD4 T cells from RRMS patients.  In the present form is not clear if there are any significant differences  in c-myc up regulation between  T cell from HD and MS.

Response: We are grateful to the reviewer for the suggestion. We did not find any significant difference in c-myc up-regulation between HD and RRMS has demonstrated by the results shown in Figure 4c. We better clarified this concept in the revised manuscript (new Figure 4c and lines 294-296).

2. Again, in Fig. 4 (panel h-m) samples from HD treated as MS samples including the treatment with anti-IL6 mAb should be added.  

Response: As required, we added the results of c-myc western blotting with the relative quantification from a representative HD in a new Supplementary Figure (Figure S4b).

3. Quantitation of IL-6 induced by anti-CD28 treatment in both HD and MS should be also measured. Is there any difference in IL-6 secretion between HD and MS under this condition?

Response: We have performed several experiments of IL-6 gene expression and secretion in CD4+ T cells isolated from either HD or RRMS (Camperio et al Immunol. Lett. 2014; Kunkl et al. Front. Immunol. 2016; Kunkl et al. Front Immunol. 2019). Thus, in the present paper we just analysed cytokine gene expression in both HD and RRMS. The results have been added in the new Supplementary Figure (Figure S4c) and no significant differences between HD and RRMS were observed.

 4. Analysis of STAT3 and NFKB activation in MS and HD following IL-6 or anti-CD28 treatment should be included in figure 4 or in a supplementary figure. 

Response: The results on STAT3 activation and NF-kB nuclear translocation in T cells from HD have been recently published (see Figure 3 and 4 of Kunkl et al. Front. Immunol 2019 doi: 10.3389/fimmu.2019.00864). Unfortunately for RRMS patients, due to the low quantity of CD4+ T cells purified from 30 cc of blood, we were not able to perform the experiments of NF-kB nuclear translocation. For this reason, we used a selective inhibitor of NF-kB, PS1145, to verify if, as observed in HD, NF-kB was involved in the expression of c-myc. The results of STAT3 tyr phosphorylation and the effects of anti-IL-6 Abs in CD4+ T cells from RRMS have been added in the new supplementary figure (Figure S4a), as required.

Round 2

Reviewer 2 Report

I am satisfied that the authors have addressed my comments.